# The Role of Cytokines in the Pathogenesis and Treatment of Alcoholic Liver Disease

**DOI:** 10.3390/diseases12040069

**Published:** 2024-03-29

**Authors:** Giuseppe Guido Maria Scarlata, Carmen Colaci, Marialaura Scarcella, Marcello Dallio, Alessandro Federico, Luigi Boccuto, Ludovico Abenavoli

**Affiliations:** 1Department of Health Sciences, University “Magna Græcia”, Viale Europa, 88100 Catanzaro, Italy; giuseppeguidomaria.scarlata@unicz.it (G.G.M.S.); carmen.colaci@studenti.unicz.it (C.C.); 2Anesthesia, Intensive Care and Nutritional Science, Azienda Ospedaliera “Santa Maria”, Via Tristano di Joannuccio, 05100 Terni, Italy; m.scarcella@aospterni.it; 3Hepatogastroenterology Division, Department of Precision Medicine, University of Campania “Luigi Vanvitelli”, Piazza Miraglia 2, 80138 Naples, Italy; marcello.dallio@unicampania.it (M.D.); alessandro.federico@unicampania.it (A.F.); 4Healthcare Genetics and Genomics Doctoral Program, School of Nursing, College of Behavioral, Social and Health Sciences, Clemson University, Clemson, SC 29634, USA; lboccut@clemson.edu

**Keywords:** alcohol, oxidative stress, inflammation, gut dysbiosis, biological drugs, probiotics

## Abstract

Alcoholic liver disease (ALD) is a major cause of chronic liver disease. This term covers a broad spectrum of liver lesions, from simple steatosis to alcoholic hepatitis and cirrhosis. The pathogenesis of ALD is multifactorial and not fully elucidated due to complex mechanisms related to direct ethanol toxicity with subsequent hepatic and systemic inflammation. The accumulation of pro-inflammatory cytokines and the reduction of anti-inflammatory cytokines promote the development and progression of ALD. To date, there are no targeted therapies to counter the progression of chronic alcohol-related liver disease and prevent acute liver failure. Corticosteroids reduce mortality by acting on the hepatic-systemic inflammation. On the other hand, several studies analyzed the effect of inhibiting pro-inflammatory cytokines and stimulating anti-inflammatory cytokines as potential therapeutic targets in ALD. This narrative review aims to clarify the role of the main cytokines involved in the pathogenesis and treatment of ALD.

## 1. Introduction

Alcoholic liver disease (ALD) is one of the leading causes of chronic liver disease worldwide. In this way, alcohol consumption is often a cofactor in patients with hepatitis B and C as well as patients with non-alcoholic fatty liver disease [1]. In this last case, an analysis of prospective studies has demonstrated how even minimal alcohol consumption has been associated with potential disease progression, particularly the onset of cancer [2]. According to the European Association for the Study of the Liver, the diagnosis of ALD should be suspected in the presence of liver damage (clinical signs and/or bio-humoral abnormalities) and a regular alcohol consumption of >30 g/day in men or >20 g/day in women [3]. ALD comprises a broad spectrum of liver lesions ranging from simple alcoholic fatty liver (AFL) to alcoholic steatohepatitis (ASH) and alcoholic hepatitis (AH) [4]. AFL diagnosis is established in patients with known alcohol use disorder (AUD) and hepatic steatosis observed on abdominal ultrasound combined with increased liver enzymes and the absence of other causes of liver disease. However, it is a serious problem due to the nonspecific symptoms [1,5]. Similar to AFL, mild ASH rarely presents with clinical symptoms and can only be diagnosed through liver biopsy. AH is a clinical condition with a high mortality rate, and it is diagnosed through the presence of jaundice in the preceding eight weeks and elevated transaminase levels. Patients with AH may present signs of severe hepatic decompensation, such as ascites and hepatic encephalopathy [6]. The pathogenesis of ALD is multifactorial due to complex molecular pathways, and the exact mechanisms are not yet fully elucidated. Certainly, a key role is played by the direct hepatotoxicity of ethanol, lipid peroxidation, oxidative stress with increased reactive oxygen species (ROS) production, and activation of the immune response via cytokines [7]. Although there are clear links between the amount and duration of alcohol consumption and the progression of ALD, other genetic and environmental factors are involved in the development and progression of the disease [8]. This narrative review aims to clarify the role of the main cytokines involved in the pathogenesis and treatment of ALD.

## 2. Epidemiology and Social Impact of ALD

About 2 million deaths each year are linked to chronic liver disease worldwide [9]. Alcohol consumption is estimated to affect 43% of the global population [10]. The prevalence of AUD stands at approximately 5%, affecting a considerable 283 million individuals globally. The European region exhibits the highest prevalence rates among both men (14.8%) and women (3.5%), followed by the America (11.5% for men and 5.1% for women) [11]. Alcohol emerges as the primary cause of liver cirrhosis on a worldwide scale, accounting for nearly 60% of cases across Europe, North America, and Latin America. Notably, approximately 35% of individuals diagnosed with AUD go on to develop various manifestations of ALD [12]. Estimates suggest a global prevalence of 23.6 million individuals living with compensated cirrhosis and 2.46 million with decompensated cirrhosis due to alcohol consumption [13]. In recent years, there has been an increase in the incidence of ALD, especially in the 15–45 age group [14]. This is a serious health problem because higher alcohol consumption in late adolescence increases the risk of developing severe liver disease, with a higher risk of death from cardiovascular diseases and cancer [15,16]. Mortality related to ALD has increased over the last decade, particularly in developed countries such as Europe, Asia, Latin America, and the United States [17,18]. In addition, another increase in ALD deaths was observed during the Coronavirus Disease-19 pandemic due to higher alcohol consumption related to emotional stress and difficult access to treatments [19]. Especially in its final stage, represented by liver cirrhosis, ALD has a high socio-economic impact. It is estimated that in 2016, liver disease-related spending in the United States was $32.5 billion. Two-thirds of these costs are attributable to hospital or emergency room care. In 20 years, healthcare spending on liver disease has increased by 4% per year [20]. Based on ALD-related disability-adjusted life years from 2016, 21.5 million life years were lost due to ALD, with men affected significantly more than women. A significant portion of these lost life years was attributable to premature death rather than disability [21]. Based on the healthcare claim analysis report, of those who survived, more than 50% were hospitalized within one year and almost 75% in the second year, with a total cost of about $145,000 for each patient [22]. Patients with AUD have an increased risk of psychiatric comorbidities, such as anxiety, affective disorders, and schizophrenia [23]. Malnutrition is frequent in ALD patients due to reduced caloric intake, abnormal digestion, increased protein catabolism, and abnormal lipid metabolism [24,25]. Patients with ALD should receive appropriate nutritional assessment and support. This has a major socio-economic impact, as in many cases, hospitalization is required for adequate parenteral or enteral nutrition [26]. In this way, folate deficiency results from reduced dietary folate intake, intestinal malabsorption, reduced hepatic absorption and deposition, and increased urinary excretion [27,28]. This event causes megaloblastic anemia as a main complication. In addition, chronic alcoholics also have thiamine, vitamin B6, and vitamin B12 deficiency [29]. Several studies conducted in animal models have shown how these deficits contribute to the progression of ALD and the development of hepatocellular carcinoma (HCC) and colorectal cancer due to an epigenetic mechanism involving DNA methylation [30,31]. On the other hand, thiamine deficiency is associated with Wernicke encephalopathy (WE), which presents with altered mental status, gait ataxia, and ophthalmoplegia. About 80% of patients with untreated WE develop Korsakoff’s syndrome (KS), characterized by memory disturbances associated with confabulation [32]. A study conducted by Wilson et al. showed that treatment with intravenous thiamine for 5 days rather than 2 days increases acute care costs but reduces the expected lifecycle cost if the patient develops KS [21]. On the other hand, a study performed by Thompson et al. defined how healthcare costs for the management of ALD increased among US patients from 2006 to 2013. Total costs were nearly USD 145,000 per patient, decreasing from USD 50,000 in the first year to USD 10,000 per year in the last years. Liver-transplanted patients averaged about USD 300,000 in transplant-related costs and over USD 1,000,000 in total healthcare costs over five years [33].

## 3. ALD Pathogenesis

The pathogenesis of ALD is based on multiple and complex molecular mechanisms that are not yet fully elucidated. These include direct ethanol hepatotoxicity, lipid peroxidation, oxidative stress with consequent ROS production, activation of the immune response, and activation of proinflammatory cytokines [34]. Ethanol metabolism comprises three pathways: (i) hepatocyte cytoplasmic alcohol dehydrogenase oxidizes ethanol into acetaldehyde, a highly toxic compound that can alter DNA synthesis; (ii) the enzyme cytochrome P450 2E1 (CYP2E1), which oxidizes ethanol into acetaldehyde, generates ROS, and triggers oxidative stress and inflammation; (iii) the heme-containing catalase found within peroxisomes facilitates the oxidation of ethanol into acetaldehyde. Following this, the enzyme aldehyde dehydrogenase (ADH) further oxidizes acetaldehyde into acetate. Acetate is then released into the circulatory system and subsequently undergoes oxidation into carbon dioxide within different tissues outside the liver [35,36,37,38]. Chronic alcohol use increases CYP2E1 expression, resulting in increased acetaldehyde concentration, decreased ADH activity, reduced acetaldehyde oxidation, and accumulation of acetaldehyde in hepatocytes. This explains the direct hepatotoxicity of ethanol on the liver [39]. In hepatocytes, ethanol, and acetaldehyde downregulate adiponectin (a peptide hormone secreted by adipose tissue, but its receptor, AdipoR2, is predominantly expressed in the liver), signal transducer and activator of transcription 3 (STAT3), and reduce zinc concentrations. This leads to the inhibition of 5′-AMP-activated protein kinase and peroxisome proliferator-activated receptor α, ultimately causing lipid peroxidation and the production of ROS [40]. The increase in ROS damages mitochondrial DNA and proteins, causing a reduction in mitochondrial ATP and glutathione. ROS-induced release of hepatocellular kinase 1 regulates the hepatocellular apoptosis signal through the cleavage of pro-caspase-3 into active caspase-3 [41]. Chronic alcohol consumption is involved in the accumulation of intestinal endotoxins and increased permeability of the intestinal wall, facilitating the translocation of lipopolysaccharide (LPS) from the gut to the liver. LPS can attach to toll-like receptors (TLRs), initiating the production and discharge of cytokines and inflammatory substances. These include tumor necrosis factor-α (TNF-α), interleukin (IL)-1β, IL-6, and platelet-derived growth factor. Consequently, this process intensifies the buildup of neutrophils and macrophages, leading to inflammation within the liver and inducing systemic harm, particularly within liver Kupffer cells [42]. Additionally, liver damage activates the proliferation of hepatic stellate cells (HSCs), which enhance the secretion of transforming growth factor-β (TGF-β) and collagen synthesis, thereby leading to fibrogenesis [43]. In recent years, the link between changes in gut microbiota composition and ALD development and progression has been investigated [44]. Several studies employing preclinical and clinical models have shown that chronic alcohol consumption induces a decrease in Bacteroidetes and Firmicutes and an increase in Enterobacteriaceae and Proteobacteria, leading to gut dysbiosis [45,46]. Gut dysbiosis is associated with increased intestinal permeability linked to altered tight junctions, with the translocation of pathogen-associated molecular patterns (PAMPs) into the liver through the portal vein [47]. In addition, alcohol abuse is associated with decreased levels of butyrate-producing genera and increased levels of proinflammatory Enterobacteriaceae. This leads to liver damage due to the PAMP translocation process [48]. The different pathogenetic pathways involved in ALD development are summarized in Figure 1.

## 4. Role of Pro-Inflammatory Cytokines

### 4.1. TNF-α

Alcohol-induced damage occurs at several levels, from innate immune cells to hepatocytes. Innate immune cells, including hepatic macrophages (Kupffer cells), play a key role in early alcohol-induced liver damage through the recognition of LPS in the portal circulation. This results in the production of LPS-induced inflammatory cytokines [49]. Among the pro-inflammatory cytokines that promote liver damage, a key role is played by TNF-α [50]. Recognition of LPS activates members of the mitogen-activated protein kinase family, including extracellular receptor-activated kinase 1/2 (ERK1/2), p38, and the c-jun-N-terminal kinase, resulting in TNF-α production, mediated through oxidative stress [51]. LPS stimulates TLR4, and at the same time, NADPH oxidase (NOX) interacts with the COOH-terminal region of TLR4, resulting in the generation of ROS in neutrophils and monocytes, which directly activates nuclear factor-κB (NF-κB), with increased TNF-α concentration [52]. A study conducted by Thakur et al. showed how the use of diphenyliodonium and dilinoleoyl-phosphatidylcholine in Kupffer cells of mice models reduced ERK1/2 activation, inhibited NOX4, resulting in lower ROS and TNF-α production [53]. Convincing evidence about the key role of TNF-α in the development of ALD was obtained from transcriptome studies that confirmed its up-regulation in patients with ALD [54]. Furthermore, a higher serum concentration correlates with a worse prognosis and an advanced disease stage [55]. Gonzales-Quintela et al. showed that serum TNF-α levels were almost similar in the general population, teetotalers, and alcohol drinkers, while in chronic alcoholics, they were elevated [56]. Other studies have shown how NF-kB activation is differently linked to chronic consumption and acute exposure to alcohol. In this regard, chronic alcohol consumption prolongs NF-kB activation, resulting in TNF-α production at the hepatic level, while acute alcohol exposure inhibits NF-kB activation, causing a reduction in TNF-α levels in the liver [57,58]. A study performed by Mookerjee et al. showed that TNF-α is an important mediator of portal and systemic hemodynamic disturbances in ALD. This was the first study in which it was observed that the use of the biological drug infliximab improved cardiovascular hemodynamics, hepatic venous pressure gradient, and hepatic and renal blood flow only twenty-four hours after its use [59]. Although the role of TNF-α in the pathogenesis of ALD is pro-inflammatory, it is interesting to note that in other conditions unrelated to the pathogenetic process of ALD, it plays a regenerative and proliferative role in hepatocytes [60]. Indeed, following hepatectomy or partial liver transplantation, bacterial LPS translocates from the intestine to the liver via the portal vein and activates Kupffer cells, which in turn produce TNF-α. Consequently, TNF-α stimulates NF-kB, resulting in the activation of downstream pathways, including PI3K/Akt/mTOR, and subsequent hepatocyte proliferation and liver regeneration [61].

### 4.2. IL-8 and CXCL1

Neutrophil infiltration is a key feature of ALD. Unlike other innate immune cells, neutrophils are not liver-resident immune cells [62]. IL-8 plays a key role in the recruitment of neutrophils to liver tissue, involving the chemokine CXC motif chemokine ligand 1 (CXCL1). Indeed, serum and liver levels of IL-8 and CXCL1 are directly correlated with disease severity and mortality [63]. A study performed by Patel et al. showed that IL-8 serum levels were significantly higher in patients with severe AH than in those with mild AH, and these levels were better predictors of short-term mortality than conventional prognostic scores [64]. Wieser et al. demonstrated how the blockade of IL-8 receptors with short lipopeptides (pepducin) in mouse models reduced liver inflammation, weight loss, and mortality associated with ALD [65]. Nischalke et al. also showed that the CXCL1 rs4074 single nucleotide polymorphism was associated with increased blood levels of CXCL1 and an increased risk of developing cirrhosis in alcoholics as well as the development of HCC [66]. Mouse models treated with a high-fat and ethanol-rich diet significantly upregulated the hepatic expression of several chemokines, including CXCL1, with a reduction of the infiltration and hepatic damage of neutrophils [67]. Roh et al. showed that the production of cytokines, including CXCL1, is mediated by TLR-2 and TLR-9, activated by LPS translocated from the gut and expressed on Kupffer cells. These data showed that in ethanol-treated wild-type mice, there was an increase in the hepatic expression of CXCL1 and the serum level of CXCL1, while TLR2- and TLR9-deficient mice showed significantly lower levels [68].

### 4.3. IL-1β

As previously reported, chronic exposure to ethanol in ALD sensitizes Kupffer cells to activation by LPS through TLR-4, inducing the production of proinflammatory cytokines, including IL-1β [69]. This is produced as a result of inflammasome activation, mainly of NOD-like receptor protein 3, highly expressed in macrophages and liver monocytes but lower expressed by hepatocytes and stellate cells [70]. NLR family pyrin domain containing 3 (NLRP3) is activated upon binding of PAMPs to TLR. This results in NF-κB activation, culminating in the transcription of pro-IL-1β precursors. Subsequently, procaspase-1 binds to NLRP3, transforming into caspase-1, which is catalytically active for IL-1β processing [71]. A study by Petrasek et al. conducted in murine models found that protein levels of NLRP3 and IL-1β were significantly increased in the liver of chronic alcohol-fed mice compared to the control group. Furthermore, data showed that mice deficient in the NLRP3 and caspase-1 inflammasome components had less steatosis and liver damage than wild-type mice on an alcoholic diet. In addition, IL-1R1 knockout mice were protected against ALD [72]. Cui et al. demonstrated, in Kupffer cells of ethanol-fed mice, an up-regulation of NLRP3 inflammasome components as well as IL-1β [73]. Similarly, a study by Voican et al. showed that increased levels of IL-1β were reported in ALD patients after one week of alcohol withdrawal [74]. In vitro experiments have shown that IL-1β is able to act directly on HSCs, with subsequent proliferation and trans-differentiation in myofibroblasts, along with an increase in collagen and TGF-β levels [75]. Another function of IL-1β is the activation of invariant natural-killer T lymphocytes (iNKT) [76]. The main role of iNKTs is to determine the hepatic infiltration of neutrophils, a hallmark in the pathogenesis of ALD [77]. iNKT cell-deficient mice were protected from hepatic infiltration of neutrophils and liver damage induced by chronic ethanol binges. In contrast, wild-type mice showed intense hepatic infiltration of neutrophils and marked upregulation of hepatic expression of several inflammation-associated genes. IL-1β also inhibits liver regeneration [78]. Finally, mice treated with an IL-1 receptor antagonist showed better regeneration of hepatocytes and an increased rate of recovery from liver damage induced by chronic ethanol consumption than untreated mice [79]. Table 1 summarizes the different studies about the involvement of pro-inflammatory cytokines in ALD.

## 5. Role of Anti-Inflammatory Cytokines

### 5.1. IL-6 and IL-10

To date, IL-6, IL-22, and IL-10 are the identified cytokines with anti-inflammatory and hepatoprotective roles in ALD. IL-6 and IL-10 regulate the expression of target genes involved in promoting cell proliferation, survival, and differentiation through STAT3 [80]. Kupffer cells express high levels of receptors for IL-10. This binding leads to prolonged activation of STAT3 and inhibits inflammatory responses. Conversely, IL-6 binding to its own receptor leads to transient activation of STAT3, followed by the induction of inflammatory responses. Activation of STAT3 induces the expression of suppressor of cytokine signaling 3, which in turn inhibits STAT3 activation by IL-6 but does not inhibit IL-10 signaling, leading to an anti-inflammatory effect [81]. Currently, the role of IL-6 in ALD is complex and not entirely clear. Based on the inflammatory trigger, certain ILs may possess both pro-inflammatory and anti-inflammatory roles. IL-6 exemplifies this duality, functioning as a pro-inflammatory cytokine in chronic inflammatory conditions while conversely demonstrating anti-inflammatory properties during acute inflammation [82]. However, few studies in the literature on this topic refer to liver cirrhosis or acute liver failure of different etiologies, with only a few patients showing such conditions associated with alcoholic etiology [83,84]. Some studies have shown how IL-6 expression can reduce apoptosis in hepatocytes and lead to mitochondrial DNA repair [85]. In this regard, a study by Zhang et al. in mouse models showed that IL-6 can activate enzymes to repair mitochondrial DNA in hepatocytes damaged by chronic alcohol consumption [86]. IL-6 promotes the differentiation of T helper 17 cells and the production of IL-17, contributing to alcohol-related inflammation. However, the activation of STAT3 by IL-6 and IL-10 inhibits the production of other proinflammatory cytokines, reducing liver damage in patients with ALD [87]. Pretreatment with IL-6 induces hepatoprotection of steatotic liver isotransplants by preventing the apoptosis of sinusoidal endothelial cells, improving hepatic microcirculation, and protecting against hepatocyte death [88]. Another study by El-Assal et al. showed that IL-6 knockout mice fed with alcohol exhibited increased liver fat accumulation, lipid peroxidation, mitochondrial DNA damage, and hepatocyte sensitization to TNFα-induced apoptosis [89]. IL-10 knockout mice showed a more severe hepatic inflammatory response with higher levels of IL-6 and STAT3 activation than wild-type mice but lower liver steatosis severity and hepatocellular damage after higher alcohol levels or fatty dietary regimen [90]. Byun et al. demonstrated how treatment with polyinosinic:polycytidylic acid in vitro stimulates IL-10 production in HSCs through TLR-3 activation with reduction of alcoholic liver damage [91]. IL-10 plays an important role in the negative regulation of liver regeneration by limiting the inflammatory response and subsequently mitigating hepatic STAT3 activation [92]. A study conducted by Yang et al. on 40 ethnically Taiwanese Han patients with alcoholic liver cirrhosis showed how certain IL-10 promoter polymorphisms allow the development of severe forms of the disease. This result clarifies how a better knowledge of the genetic background of alcoholic liver cirrhosis is essential for its prevention and treatment [93].

### 5.2. IL-22

IL-22 is a cytokine belonging to the IL-10 family due to the similarity in genetic and protein structures. It is involved in several diseases, such as inflammatory bowel diseases and skin, pancreatic, lung, and liver diseases [94]. IL-22 is only produced by hematopoietic cells, while the interleukin-22 receptor (IL-22R) is expressed in different organs. It is a heterodimer composed of IL-10R2 and IL-22R1; while IL-10R2 is a subunit shared with various IL-10 family receptor complexes, IL-22R1 is only expressed in cells of various organs such as bronchi, liver, pancreas, and gut [95]. IL-22 plays a protective role in alcoholic liver damage through the activation of the STAT3-mediated signaling pathway. This event promotes the overexpression of anti-apoptotic and mitogenic genes, leading to tissue repair and survival of hepatocytes [96]. A recent study performed in mouse models with ALD showed that four weeks of treatment with IL-22 reduced autophagy and liver fibrosis in hepatocytes [97]. In a recent prospective cohort study, IL-22 levels were significantly increased in alcohol-associated non-severe hepatitis and alcohol-associated severe hepatitis patients compared to the control group. As reported by the Authors, increased response to proinflammatory cytokines can mitigate liver damage [98]. Another investigation has clarified the role of IL-22 in liver regeneration. It was observed how IL-22 was significantly increased in murine models following liver regeneration and how this signaling blockade caused a reduction in liver regenerative capacity [99]. Table 2 summarizes the different studies on the involvement of anti-inflammatory cytokines in ALD. As previously reported, chronic alcohol consumption causes gut dysbiosis. Indeed, gut microflora imbalance reduces IL-22 production in the small intestine in mice models, which in turn leads to a decrease in antimicrobial C-type lectin regenerating islet-derived 3 gamma (REG3G). The reduction of REG3G causes an increase in bacterial translocation to the liver with consequent steatohepatitis [100].

## 6. New Therapeutic Approaches

### 6.1. Biological Drugs

The treatment of ALD is based on AUD therapy and the management of severe hepatitis associated with alcohol abuse [101]. To date, the drugs approved for AUD are disulfiram, naltrexone, nalmefene, and acamprosate. Disulfiram, naltrexone, and acamprosate are used for alcohol withdrawal, while nalmefene acts to reduce alcohol consumption. These drugs act on ethanol metabolism or, at the central nervous system level, by modulating the opioid pathway [102]. Two other drugs, sodium oxybate and baclofen, are used to treat AUD in some countries, such as Italy and Austria [103]. To date, treatment for severe AH benefits exclusively from the use of corticosteroids. The basic mechanism is the reduction of inflammation through cytokine modulation [104]. Corticosteroids increase survival in the short term but not in the long term and are associated with collateral effects, such as infections [105,106]. Since ALD is characterized by a cytokine storm, a potential treatment may be the inhibition of pro-inflammatory cytokines or the increase in anti-inflammatory cytokines levels. In this way, pentoxifylline, a selective phosphodiesterase inhibitor, has been tested as a therapeutic agent for the treatment of severe AH, acting with a reduction of pro-inflammatory cytokines levels [107]. The Steroids or Pentoxifylline for Alcoholic Hepatitis (STOPAH) study comparing the efficacy of pentoxifylline and prednisolone showed that pentoxifylline did not improve survival in AH patients, while prednisolone reduced 28-day mortality [108]. Other agents, such as infliximab and etanercept, which act by blocking TNF-α production, have been studied for the treatment of severe AH. A study conducted in patients with severe AH treated with infliximab and prednisolone showed an increased mortality rate compared to patients treated with prednisolone alone [109]. A further study clarified how a single dose of infliximab could improve the severity and survival rate in severe AH. However, in both studies, the most common side effect was the development of infections. For this reason, further investigations are needed in order to use them as a potential therapeutic target [110,111]. Similarly to infliximab, etanercept has also proved unsuccessful in the treatment of severe AH [112]. Recently, new therapeutic targets have emerged. Already in preclinical models, IL-22 has proven effective in reducing liver damage associated with ALD [113,114]. Phase I clinical trials have demonstrated the safety and efficacy of recombinant human IL-22 Immunoglobulin G (IgG)2-Fc, showing that the latter has good tolerance with good pharmacokinetic and pharmacodynamic properties at the intravenous dose of 45 μg/kg [115,116]. Recently, an open-label phase II study was conducted on F-652, a recombinant fusion protein of human IL-22 and IgG2-Fc. Specifically, 18 patients (9 with moderate AH and 9 with severe AH) were enrolled, and three growing doses of F-652 (10, 30, and 45 μg/kg) were administered. The treatment proved to be safe, and the patients showed a high rate of improvement as determined via the Lille and Model for End-Stage Liver Disease prognostic scores, with decreased levels of inflammatory biomarkers and increased levels of liver regeneration biomarkers compared to control subjects. Despite this, the small sample number defines how multicenter, randomized, placebo-controlled studies are needed to confirm the benefits of IL-22Fc therapy in patients with moderate-severe AH [117]. IL-22 has been shown to promote the development of HCC via a STAT3-mediated mechanism [118]. However, studies performed on transgenic mouse models with high levels of IL-22 showed no higher incidence of spontaneous tumor development than in wild-type mice [96]. Another therapeutic target for the treatment of severe AH may be IL-1β inhibition. The ongoing IL-1 Signal Inhibition in Alcoholic Hepatitis (ISAIAH) study evaluates the efficacy of the monoclonal antibody canakinumab at a dose of 3 mg/kg intravenously at baseline and after 28 days. The primary endpoint obtained was the histological improvement of AH on liver biopsy after 28 days of treatment compared to baseline, defined as a reduction in lobular inflammation [119]. A study was also conducted on the use of anakinra, an interleukin-1β receptor antagonist. Therapy with anakinra 100 mg via subcutaneous injection for 14 days plus zinc sulfate 220 mg for 90 days was administered and compared to prednisone 40 mg PO daily for 30 days. As reported by the Authors, therapy with anakinra plus zinc sulfate improved survival by 90 days in patients with severe AH [120]. However, evaluating if the drug can be eliminated efficiently in patients with impaired kidney function [117] is necessary. Overall, the abovementioned study provided better knowledge about the effects and metabolism of anakinra in patients with severe AH [121]. Table 3 summarizes the different mechanisms of action of the drug treatments.

### 6.2. Gut Microbiota Modulation

Gut microbiota modulation can be a safe therapeutic strategy for improving patients’ quality of life [122]. In this way, a recent study performed on mouse models with alcoholic liver damage showed that the probiotic *Lactobacillus plantarum* reduces the abundance of Gram-negative bacteria, with a reduction of the LPS content in the gut. Moreover, *Lactobacillus plantarum* J26 is able to maintain the integrity of the gut barrier, preventing bacterial translocation to the liver with a consequent reduction of the inflammation linked to alcohol consumption [123]. Another study on murine models compared the protective role of two strains of *Lactobacillus plantarum*, E680 and ZY08, in the ALD. Data reported that the intervention with *Lactobacillus plantarum* ZY08 significantly mitigated alcohol-related hepatic steatosis, liver damage and gut dysbiosis and relieved plasma LPS levels as well as liver lipid metabolism [124]. A recent study performed in patients with severe AH showed that daily oral administration of *Lactobacillus rhamnosus* GG is associated with a significant reduction in liver damage after one month of treatment [125]. A double-blind, randomized, placebo-controlled, multicenter study is still ongoing with the aim of evaluating the efficacy of bovine colostrum in the treatment of AH. Bovine colostrum has two important components, lactoferrin and IgG: lactoferrin binds to lipid A to neutralize it, and IgG interacts with lymphoid tissue associated with the mucosa and reduces intestinal permeability. Indeed, this compound acts as an immunomodulator, reducing the inflammation typical of ALD [126]. Overall, further studies are required to better define the key role of gut microbiota modulation in the treatment of liver diseases [127,128].

## 7. Conclusions

In this narrative review, we evaluated the cytokines mainly implicated in the development of ALD and those that have emerged as potential therapeutic targets. However, additional pro-inflammatory and anti-inflammatory cytokines play a key role in the pathogenesis and progression of ALD. For this reason, clarifications about the immune pathways involved in this pathology are needed. At the same time, new therapeutic approaches, such as biological drugs and probiotics as targets of cytokines and the gut microbiota that modulates them, maybe a good therapeutic option. Despite this, new studies are needed to evaluate the use of these promising therapies and their beneficial and adverse effects on ALD.

## Figures and Tables

**Figure 1 diseases-12-00069-f001:**
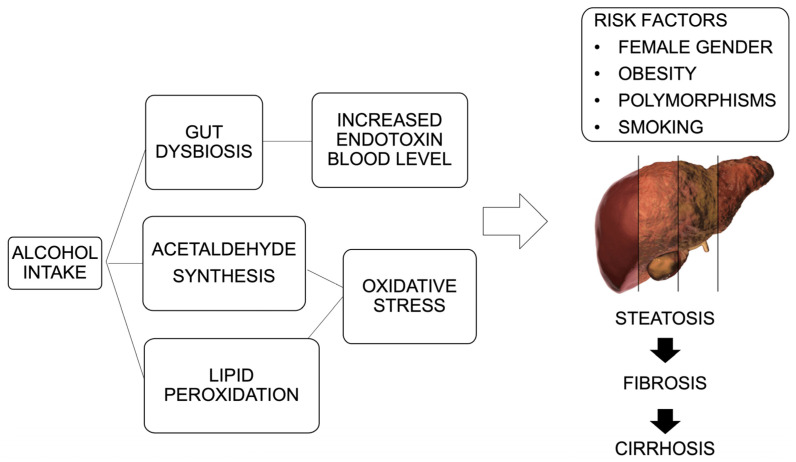
Schematic representation of the different pathogenetic ways and risk factors involved in ALD development.

**Table 1 diseases-12-00069-t001:** Summary of different studies about the involvement of pro-inflammatory cytokines in ALD.

Cytokine Evaluated	Reference	Study Design	Outcome
TNF-α	Thakur et al., 2006 [53]	Pre-clinical study	Reduction of ERK1/2 activation and inhibition of NOX4 resulting in lower ROS and TNF-α production after the use of diphenyl-iodonium and dilinoleoyl-phosphatidylcholine in Kupffer cells of mouse models
	Ciećko-Michalska et al., 2006 [55]	Case-control study	Higher concentrations of TNF-α in ALD patients were correlated with poor prognosis
	Mandrekar et al., 2006 [58]	Pre-clinical study	Significant reduction in monocyte production of TNF-α in response to LPS or staphylococcal enterotoxin B stimulation eighteen hours after moderate alcohol consumption
	Gonzales-Quintela et al., 2008 [56]	Case-control study	TNF-α levels were almost similar in the general population, teetotalers, and alcohol drinkers, while elevated in chronic alcoholics
	Affò et al., 2013 [54]	Translational study	TNF superfamily receptors are overexpressed in AH humans and mice models
IL-8	Patel et al., 2015 [64]	Case-control study	Serum IL-8 predicts severity and mortality in patients with AH
	Wieser et al., 2017 [65]	Pre-clinical study	Blockade of IL-8 receptors with pepducin reduced liver inflammation, weight loss, and mortality associated with ALD in mice models
CXCL1	Nischalke et al., 2013 [66]	Case-control study	CXCL1 rs4074 single nucleotide polymorphism was associated with increased blood levels of CXCL1 and an increased risk of developing liver cirrhosis and HCC
	Chang et al., 2015 [67]	Pre-clinical study	Mouse models treated with a high-fat and ethanol-rich diet markedly upregulated the hepatic expression of CXCL1, with a reduction of the infiltration and hepatic damage of neutrophils
	Roh et al., 2015 [68]	Pre-clinical study	Ethanol-treated wild-type mice showed an increased hepatic expression of CXCL1 and serum level of CXCL1, while TLR2- and TLR9-deficient mice showed significantly lower levels
IL-1β	Petrasek et al., 2012 [72]	Pre-clinical study	IL-1β levels were significantly increased in the liver of chronic alcohol-fed mice compared to controls
	Cui et al., 2015 [73]	Pre-clinical study	Up-regulation of NLRP3 inflammasome components and IL-1β in Kupffer cells of ethanol-fed mice
	Voican et al., 2015 [74]	Prospective cohort study	Significantly increased levels of IL-1β in ALD patients after one week of alcohol withdrawal
	Mathews et al., 2016 [78]	Pre-clinical study	A loss of function of IL-1β protected mouse models from hepatic infiltration of neutrophils and liver damage induced by chronic ethanol binges

Legend: TNF-α, tumor necrosis factor-alpha; ERK1/2, extracellular signal-regulated kinase 1/2; NOX4, NADPH oxidase 4; ROS, reactive oxygen species; ALD, alcoholic liver disease; LPS, lipopolysaccharide; AH, alcoholic hepatitis; IL-8, interleukin-8; CXCL1, CXC motif chemokine ligand 1; TLR2, toll-like receptor 2; TLR9, toll-like receptor 9; IL-1β, interleukin-1 beta; NLRP3, NLR family pyrin domain containing 3.

**Table 2 diseases-12-00069-t002:** Summary of different studies about the involvement of anti-inflammatory cytokines in ALD.

Cytokine Evaluated	Reference	Study Design	Outcome
IL-6	Sun et al., 2003 [88]	Pre-clinical study	Pretreatment with IL-6 induces hepatoprotection of steatotic liver isotransplants by preventing the apoptosis of sinusoidal endothelial cells, improving hepatic microcirculation, and protecting against hepatocyte death in mouse models
	El-Assal et al., 2004 [89]	Pre-clinical study	IL-6 knockout mice fed with alcohol exhibited increased liver fat accumulation, lipid peroxidation, mitochondrial DNA damage, and hepatocyte sensitization to TNFα-induced apoptosis
	Zhang et al., 2010 [86]	Pre-clinical study	IL-6 activates enzymes to repair mitochondrial DNA in hepatocytes damaged via chronic alcohol consumption in mouse models
	Miller et al., 2011 [90]	Pre-clinical study	IL-10 knockout mice showed a more severe hepatic inflammatory response with higher levels of IL-6 and STAT3 activation, compared to wild-type mice, but lower liver steatosis severity and hepatocellular damage after higher alcohol levels or fatty dietary regimen
IL-10	Byun et al., 2013 [91]	Pre-clinical study	Treatment with polyinosinic:polycytidylic acid in vitro stimulates IL-10 production in HSCs through TLR-3 activation with reduction of alcoholic liver damage
	Yang et al., 2014 [93]	Case-control study	IL-10 promoter polymorphisms allow the development of severe forms of alcoholic liver cirrhosis
IL-22	Meng et al., 2023 [97]	Pre-clinical study	Four-week treatment with IL-22 reduced liver fibrosis in mouse models with ALD
	Sagaram et al., 2023 [98]	Prospective cohort study	IL-22 levels were significantly increased in alcohol-associated non-severe hepatitis and alcohol-associated severe hepatitis patients compared to controls
	Liu et al., 2023 [99]	Pre-clinical study	IL-22 levels were significantly increased in murine models following liver regeneration

Legend: IL-6, interleukin-6; TNF-α, tumor necrosis factor-alpha; IL-10, interleukin-10; STAT3, signal transducer and activator of transcription 3; ALD, alcoholic liver disease; HSCs, hepatic stellate cells; TLR3, toll-like receptor 3; IL-22, interleukin-22.

**Table 3 diseases-12-00069-t003:** Summary of the different mechanisms of action of the drug treatments.

Drug Treatment	Mechanism of Action
Disulfiram	ADH antagonist
Naltrexone	Opioid receptor antagonist
Nalmefene	Opioid receptor antagonist
Acamprosate	NMDA receptor modulator
Sodium oxybate	GABA-B receptor agonist
Baclofen	GABA-B receptor agonist
Pentoxifylline	TNF-α antagonist
Infliximab	TNF-α antagonist
Etanercept	TNF-α antagonist
F-652	STAT3 activator
Canakinumab	IL-1β antagonist
Anakinra	IL-1β antagonist

Legend: ADH, aldehyde dehydrogenase; NMDA, N-methyl-D-aspartate; GABA, gamma-aminobutyric acid; TNF-α, tumor necrosis factor-alpha; STAT3, signal transducer and activator of transcription 3; IL-1β, interleukin-1 beta.

## Data Availability

Not applicable.

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
