# Peer review of "The Role of Cytokines in the Pathogenesis and Treatment of Alcoholic Liver Disease"

_diseases, 2024, doi:10.3390/diseases12040069_

Round 1

Reviewer 1 Report

Comments and Suggestions for Authors

Manuscript title: The role of cytokines in the pathogenesis and treatment of

alcoholic liver disease

Journal: Diseases

Manuscript ID: diseases-2916986

In this study, 

Scarlata et al. aimed to review the role of cytokines in the pathogenesis and treatment of alcoholic liver disease (ALD). To this end, the authors analyzed previous investigations related to the effects of pro-inflammatory anti-inflammatory cytokines on the development and progression of ALD. They presented the results as a narrative review and provided some therapeutic procedures to alleviate the consequences of ALD. The study concluded that involvement of pro- and anti-inflammatory cytokines is established but the mechanisms of action are still not defined. This review is well-presented and provides abundant information about ALD and cytokines. This field of research is interesting and highly significant to current worldwide health issues. There are some questions and concerns that need to be addressed to improve the manuscript. 

1- Lanes 121: The authors indicate “In addition, ethanol and acetaldehyde upregulate adiponectin, signal transducer and activator of transcription”. It is not clear if this effect is in the liver or located other organs since adiponectin is primarily expressed in adipose tissue. The authors need to clarify if these changes are in the liver or other tissues. Similarly, other statements are also reported throughout the manuscript without specification if it is intrahepatic or extrahepatic effects.

2- Lane 174: “Acute alcohol exposure inhibits NF-κB activation, leading to a reduction in TNF-a levels”. First, there is no indication where this effect is occurring? In the liver or another organ? Second, this statement is this statement seems contradictory previous sentences in the same paragraph statement that alcohol activates NF-κB and TNF-a. 

3- Lane 180: “TNF-a plays a key role in hepatocyte proliferation and liver regeneration”. This statement is through in without and explanation of the mechanism. In addition, this effect seems to be beneficial for the liver. How to reconcile between this effect and the pro-inflammatory effect of TNF-a reported earlier?

4- Lane 244: There is a big discussion on whether IL-6 is anti- or pro-inflammatory cytokine. In this review, the authors classify IL-6 as anti-inflammatory cytokine at the same level as IL-10. Is there consensus that IL-6 is beneficial in ALD? If not, the authors should provide both pro- and anti- investigations on this issue.

5- There are many cytokines, why the authors focused on some not the rest of them. Does this mean there is a gap in literature? It these cytokines are the most important, then simple justification could resolve this question.

6- Lane 309: It well be helpful to the readers to present a table that summarizes the effects and mechanisms of action of drugs.

7- There are some abbreviations throughout the text that need to be defined; STOPAH, ISAIAH, e.g.

Comments on the Quality of English Language

Author Response

In this study,

Scarlata et al. aimed to review the role of cytokines in the pathogenesis and treatment of alcoholic liver disease (ALD). To this end, the authors analyzed previous investigations related to the effects of pro-inflammatory anti-inflammatory cytokines on the development and progression of ALD. They presented the results as a narrative review and provided some therapeutic procedures to alleviate the consequences of ALD. The study concluded that involvement of pro- and anti-inflammatory cytokines is established but the mechanisms of action are still not defined. This review is well-presented and provides abundant information about ALD and cytokines. This field of research is interesting and highly significant to current worldwide health issues. There are some questions and concerns that need to be addressed to improve the manuscript.

1- Lanes 121: The authors indicate “In addition, ethanol and acetaldehyde upregulate adiponectin, signal transducer and activator of transcription”. It is not clear if this effect is in the liver or located other organs since adiponectin is primarily expressed in adipose tissue. The authors need to clarify if these changes are in the liver or other tissues. Similarly, other statements are also reported throughout the manuscript without specification if it is intrahepatic or extrahepatic effects.

Reply 1. Thank you for your comment. The text has been revised (see lines 126-132).

2- Lane 174: “Acute alcohol exposure inhibits NF-κB activation, leading to a reduction in TNF-a levels”. First, there is no indication where this effect is occurring? In the liver or another organ? Second, this statement is this statement seems contradictory previous sentences in the same paragraph statement that alcohol activates NF-κB and TNF-a.

Reply 2. Thank you for your comment. The text has been revised (see lines 180-184).

3- Lane 180: “TNF-a plays a key role in hepatocyte proliferation and liver regeneration”. This statement is through in without and explanation of the mechanism. In addition, this effect seems to be beneficial for the liver. How to reconcile between this effect and the pro-inflammatory effect of TNF-a reported earlier?

Reply 3. Thank you for your comment. The text has been revised (see lines 188-195).

4- Lane 244: There is a big discussion on whether IL-6 is anti- or pro-inflammatory cytokine. In this review, the authors classify IL-6 as anti-inflammatory cytokine at the same level as IL-10. Is there consensus that IL-6 is beneficial in ALD? If not, the authors should provide both pro- and anti- investigations on this issue.

Reply 4. Thank you for your comment. The text has been revised (see lines 265-271).

5- There are many cytokines, why the authors focused on some not the rest of them. Does this mean there is a gap in literature? It these cytokines are the most important, then simple justification could resolve this question.

Reply 5. Thank you for your comment. The text has been revised (see lines 28-29, 56-57, 411-414).

6- Lane 309: It well be helpful to the readers to present a table that summarizes the effects and mechanisms of action of drugs.

Reply 6. Thank you for your comment. Table 3 has been drawn (see lines 383-388).

7- There are some abbreviations throughout the text that need to be defined; STOPAH, ISAIAH, e.g.

Reply 7. Thank you for your comment. The text has been revised (see lines 299, 341-342, 371-372, 442, 447, 447, 449, 454, 460).

Reviewer 2 Report

Comments and Suggestions for Authors

The manuscript entitled “The role of cytokines in the pathogenesis and treatment of alcoholic liver disease” provides an excellent overview on the development of ALD and the possible therapies by targeting some specific cytokines. However, some remarks might be taken into consideration in order to enhance the quality of your paper.

Abstract : “To date, there are no targeted therapies to counteract disease progression and prevent acute liver failure”. Please do you mean acute or chronic liver disease?

Line 33: What do you mean by “alcohol consumption is often a cofactor in patients with non-alcoholic fatty liver disease”.

Line 47: please change this “The pathogenesis of ALD is multifactorial due to complex molecular pathways not yet fully elucidated” To The pathogenesis of ALD is multifactorial due to complex molecular pathways and the exact mechanisms are not yet fully elucidated”.

Line 49: stress with reactive oxygen species (ROS) production, to oxidative stress with increased reactive oxygen species (ROS) production

Line 52: please change “the development and progression of ALD” to the development and progression of the disease (to avoid repetition).  

Please add reference here “alcohol consumption is estimated at 43% of the global population”. Reff

Please add other factors that contribute to the development of ALD in Figure 1; and add the 3 stages of ALD: Steatosis (fatty liver disease), fibrosis, and cirrhosis.

Comments on the Quality of English Language

Minor English proofreading is highly recommended

Author Response

The manuscript entitled “The role of cytokines in the pathogenesis and treatment of alcoholic liver disease” provides an excellent overview on the development of ALD and the possible therapies by targeting some specific cytokines. However, some remarks might be taken into consideration in order to enhance the quality of your paper.

  1. Abstract: “To date, there are no targeted therapies to counteract disease progression and prevent acute liver failure”. Please do you mean acute or chronic liver disease?

Reply 1. Thank you for your comment. The abstract has been revised (see lines 24-25).

  1. Line 33: What do you mean by “alcohol consumption is often a cofactor in patients with non-alcoholic fatty liver disease”.

Reply 2. Thank you for your comment. The text has been revised (see lines 34-36).

  1. Line 47: please change this “The pathogenesis of ALD is multifactorial due to complex molecular pathways not yet fully elucidated” To “The pathogenesis of ALD is multifactorial due to complex molecular pathways and the exact mechanisms are not yet fully elucidated”.

Reply 3. Thank you for your comment. The text has been revised (see lines 49-51).

  1. Line 49: stress with reactive oxygen species (ROS) production, to oxidative stress with increased reactive oxygen species (ROS) production

Reply 4. Thank you for your comment. The text has been revised (see lines 52-53).

  1. Line 52: please change “the development and progression of ALD” to the development and progression of the disease (to avoid repetition).

Reply 5. Thank you for your comment. The text has been revised (see lines 55-56).

  1. Please add reference here “alcohol consumption is estimated at 43% of the global population”. Reff

Reply 6. Thank you for your comment. The text has been revised (see line 60).

  1. Please add other factors that contribute to the development of ALD in Figure 1; and add the 3 stages of ALD: Steatosis (fatty liver disease), fibrosis, and cirrhosis.

Reply 7. Thank you for your comment. Figure 1 has been revised (see lines 156-158).